

# Shoulder muscle weakness and proprioceptive impairments in type 2 diabetes mellitus: exploring correlations for improved clinical management

Ravi Shankar Reddy[1], Mastour Saeed Alshahrani[1], Mohammad A. ALMohiza[2], Batool Abdulelah Alkhamis[1], Jaya Shanker Tedla[1], Venkata Nagaraj Kakaraparthi[1], Ghada Mohamed Koura[1], Debjani Mukherjee[1], Hani Hassan Al-nakhli[1], Hussain Saleh H. Ghulam[3] and Raee S. Alqhtani[3]

[1] Medical Rehabilitation Sciences, King Khalid University, Abha, Aseer, Saudi Arabia
[2] Department of Rehabilitation Sciences, College of Applied Medical Sciences, King Saud University, Riyadh, Riyadh, Saudi Arabia
[3] Department of Physical Therapy, Najran University, Najran, Saudi Arabia

Corresponding author
Ravi Shankar Reddy,
rshankar@kku.edu.sa

## ABSTRACT

**Background.** Type 2 diabetes mellitus (T2DM) is a prevalent metabolic disorder with systemic implications, potentially affecting musculoskeletal health. This study aimed to assess shoulder muscle strength and joint repositioning accuracy in individuals with T2DM, exploring potential correlations and shedding light on the musculoskeletal consequences of the condition. The objectives were two-fold: (1) to assess and compare shoulder strength and joint repositioning accuracy between individuals with T2DM and asymptomatic counterparts, and (2) to examine the correlation between shoulder strength and joint repositioning accuracy in individuals with T2DM.

**Methods.** A cross-sectional study enrolled 172 participants using the convenience sampling method, including 86 individuals with T2DM and an age-matched asymptomatic group ($n = 86$). Shoulder strength was assessed using a handheld dynamometer, while joint repositioning accuracy was evaluated with an electronic digital inclinometer.

**Results.** Individuals with T2DM exhibited reduced shoulder muscle strength compared to asymptomatic individuals ($p < 0.001$). Additionally, joint repositioning accuracy was significantly lower in the T2DM group ($p < 0.001$). Negative correlations were observed between shoulder strength and joint repositioning accuracy in various directions (ranging from $-0.29$ to $-0.46$, $p < 0.001$), indicating that higher muscle strength was associated with improved joint repositioning accuracy in individuals with T2DM.

**Conclusion.** This study highlights the significant impact of T2DM on shoulder muscle strength and joint repositioning accuracy. Reduced strength and impaired accuracy are evident in individuals with T2DM, emphasizing the importance of addressing musculoskeletal aspects in diabetes management. The negative correlations suggest that enhancing shoulder muscle strength may lead to improved joint repositioning accuracy, potentially contributing to enhanced physical functioning in this population.

## INTRODUCTION

Type 2 diabetes mellitus (T2DM) is a complex metabolic disorder characterized by chronic hyperglycemia resulting from insulin resistance and inadequate insulin secretion (*Galicia-Garcia et al., 2020*). It is a significant global health concern, affecting millions of individuals worldwide and posing a substantial burden on healthcare systems (*Galicia-Garcia et al., 2020*). While T2DM primarily impacts glucose metabolism, its systemic effects can extend to various physiological systems, including musculoskeletal health (*Thyfault & Bergouignan, 2020*). One intriguing aspect of this interplay is the potential impact of T2DM on shoulder strength and joint repositioning accuracy, both of which are crucial for maintaining functional independence and overall quality of life (*Reddy & Tan, 2020*).

Research over the years has highlighted the significant impact of T2DM on musculoskeletal health, a system encompassing bones, muscles, joints, and connective tissues, all of which play critical roles in maintaining mobility and overall physical function (*Meem, 2022*). T2DM can affect these components through a variety of mechanisms (*Meem, 2022*). Skeletal muscles, responsible for generating force and facilitating movement, are directly impacted by T2DM (*Al-Ozairi et al., 2021*). Insulin resistance, a hallmark of T2DM, can lead to reduced muscle glucose uptake, potentially resulting in muscle atrophy and weakness (*Al-Ozairi et al., 2021*). Muscle strength is a fundamental aspect of physical function, influencing activities of daily living, mobility, and overall quality of life (*Al-Ozairi et al., 2021*). The joint repositioning sense, often referred to as proprioception, is vital for joint stability and coordination (*Proske & Chen, 2021*). Proprioception relies on sensory input from muscles, tendons, and ligaments, allowing individuals to perceive joint position and make rapid, precise adjustments in response to changes in posture or movement (*Gelener, Iyigün & Özmanevra, 2021*). In T2DM, alterations in blood glucose levels may impact nerve function and potentially disrupt proprioceptive feedback, which could, in turn, affect joint repositioning accuracy (*Paggi et al., 2021*; *Reddy et al., 2023*).

Several factors, including age, gender, HbA1c levels, and the duration of diabetes, can influence the musculoskeletal health of individuals with T2DM (*Sit et al., 2022*). Aging is associated with natural declines in muscle mass and strength and gender differences in musculoskeletal characteristics are well-documented (*Granic et al., 2023*). HbA1c levels, a marker of long-term glucose control, can reflect the severity of diabetes, while the duration of diabetes may indicate cumulative metabolic stress on the musculoskeletal system (*Tan, 2020*). Given the potential impact of T2DM on shoulder strength and joint repositioning accuracy, exploring these parameters in depth is of great significance (*Tan, 2020*). Reduced shoulder strength can compromise individuals' ability to perform daily tasks, affecting their independence and overall well-being (*Rodrigues et al., 2020*). Likewise, impaired joint repositioning accuracy may lead to a heightened risk of falls and injuries, particularly in older adults with T2DM, thus exacerbating the already elevated risk of complications associated with this condition (*Bodrucky, 2020*).

Understanding the potential differences and correlations between these musculoskeletal parameters in T2DM compared to asymptomatic individuals can provide valuable insights for both clinical practice and research (*ALMohiza et al., 2023*; *Faletra et al., 2022*). Such

insights could inform the development of tailored interventions to mitigate musculoskeletal complications in individuals with T2DM, potentially improving their overall health outcomes and quality of life (*Sugandh et al., 2023*). Moreover, investigating the correlations between shoulder strength and joint repositioning accuracy within the T2DM population can contribute to a more comprehensive understanding of the mechanisms underlying musculoskeletal dysfunction in this context (*Singh et al., 2023*). This knowledge may uncover specific areas for targeted rehabilitation and therapeutic strategies to enhance musculoskeletal function and, by extension, the overall well-being of individuals with T2DM.

This study has two primary aims. Firstly, it seeks to assess and compare the levels of shoulder strength and joint repositioning accuracy in individuals diagnosed with T2DM and asymptomatic individuals. Additionally, the study aims to examine the correlation between shoulder strength and joint repositioning accuracy within the T2DM group, aiming to quantify the strength of these relationships. The alternative hypotheses for the study are as follows: Firstly, there is a significant difference in shoulder strength and joint repositioning accuracy between individuals diagnosed with T2DM and asymptomatic individuals. This suggests that T2DM may have an impact on muscle strength and proprioceptive abilities. Secondly, within the T2DM group, there is a significant correlation between shoulder strength and joint repositioning accuracy. This implies that in individuals with T2DM, variations in shoulder strength could be associated with changes in the accuracy of joint repositioning, indicating a possible interdependence between muscular strength and proprioceptive function specific to this condition.

## MATERIALS & METHODS

### Study design

The cross-sectional study was conducted at a tertiary healthcare facility in Abha city, Saudi Arabia over the period spanning from March 2022 to February 2023. Stringent ethical considerations were central to the study's design, with a steadfast commitment to adhering to the ethical principles articulated in the Declaration of Helsinki. The necessary ethical clearance was obtained from the King Khalid University Ethics Committee with approval number ECM# 2022-1136, ensuring that the study adhered to established ethical standards. Additionally, all participants provided written informed consent.

### Participants

Participants for the study were selected using a convenience sampling method. This approach was chosen due to its practicality and efficiency in gathering a sample from a population with T2DM and asymptomatic individuals available at local community health centers. Convenience sampling allowed for the rapid collection of data while still providing valuable insights into the effects of T2DM on musculoskeletal health. Participants were recruited through advertisements placed in local community centers, social media, and local newspapers. Initial screening was conducted *via* telephone to verify eligibility based on predefined inclusion and exclusion criteria. A total of 164 participants, encompassing both individuals diagnosed with type 2 diabetes mellitus (T2DM) and age-matched, healthy
asymptomatic individuals, were recruited following a comprehensive initial assessment and strict adherence to predefined criteria. In the case of T2DM patients, inclusion criteria necessitated a confirmed diagnosis based on established clinical parameters, such as fasting plasma glucose levels $\geq$126 mg/dL, hemoglobin A1c (HbA1c) levels $\geq$6.5%, or a 2-hour oral glucose tolerance test result of $\geq$200 mg/dL (*Thornton-Swan et al., 2022*). Age specifications were also applied when the study aimed to investigate diabetes in specific demographic groups. Furthermore, asymptomatic status constituted a crucial inclusion criterion, ensuring that individuals included in the study did not manifest diabetes-related symptoms such as polyuria, polydipsia, unexplained weight loss, or diabetic ketoacidosis at the time of recruitment. Obtaining written informed consent and a commitment to adhere to study requirements, including medication compliance and follow-up visits, were also fundamental inclusion criteria.

Conversely, exclusion criteria for asymptomatic individuals with T2DM were implemented to refine the study population further. Symptomatic diabetes, characterized by the presence of symptoms such as polyuria, polydipsia, unexplained weight loss, or diabetic ketoacidosis, typically served as a primary exclusion criterion. Gestational diabetes, a distinct condition requiring specialized care, was also excluded. Participants with severe comorbidities, including end-stage renal disease, advanced cardiovascular disease, or other conditions that could significantly influence study outcomes or pose a risk to their health during the study, were generally ineligible for participation. Additionally, individuals with severe cognitive impairments that might impede their capacity to provide written informed consent and adhere to study requirements were excluded. Lastly, individuals with non-type 2 diabetes, such as type 1 diabetes mellitus or other specific diabetes types, were also excluded due to differing pathophysiology.

Data collection involved standardized assessments of shoulder muscle strength and proprioception using a handheld dynamometer and a digital inclinometer, respectively. All assessments were conducted by trained medical professionals in a controlled clinical setting. Additionally, structured interviews were conducted to collect qualitative data on participants' experiences and the functional impacts of T2DM.

## Isometric shoulder strength testing

Isometric shoulder strength assessments were conducted using a Handheld Dynamometer (MicroFET 2; Hoogan Health Industries, West Jordan, UT, USA) to ensure precise measurements (*Biasini et al., 2023*; *Cools et al., 2016*; *Saccol, Santos & Oliano, 2017*). Participants maintained a supine throughout the testing procedures. The strength assessments were conducted at predefined joint positions, ensuring the appropriate placement of the dynamometer (*Clarke et al., 2011*). The muscle strength of the shoulder was assessed using a standardized protocol detailed in Table 1 and Fig. 1, focusing on muscles responsible for shoulder flexion, extension, abduction, and both internal and external rotation. Participants were positioned supine for all tests: for shoulder flexion and extension, the arm was positioned at 90° of shoulder flexion, elbow flexed to 5° to 10° and with the dynamometer placed just proximal to the elbow; for shoulder abduction, the arm was abducted to 90°, also with the dynamometer just proximal to the elbow; and for internal

**Table 1** Standardized shoulder strength assessment protocol using a dynamometer.

| Muscle Group | Participant Position | Upper Limb Position | Dynamometer position |
| --- | --- | --- | --- |
| Shoulder Flexion | Supine | 90° of flexion | Just proximal to the elbow. |
| Shoulder Extension | Supine | 90° of flexion | Just proximal to the elbow. |
| Shoulder Abduction | Supine | Abducted to 90° | Just proximal to the elbow. |
| Shoulder Internal Rotation | Supine | Shoulder abducted to 45° and elbow flexed to 90° | Just proximal to the wrist |
| Shoulder External Rotation | Supine | Shoulder abducted to 45° and elbow flexed to 90° | Just proximal to the wrist |

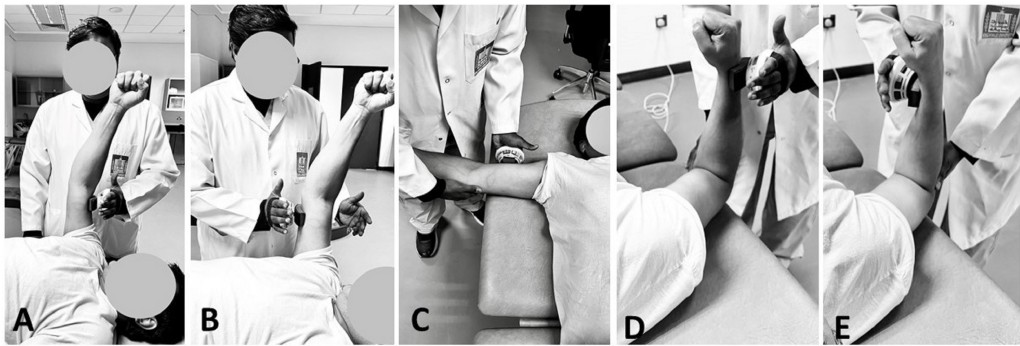

**Figure 1** Isometric Shoulder Strength Testing using hand-held dynamometer in (A) flexion; (B) extension; (C) abduction; (D) external rotation and (E) internal rotation.

and external rotation, the shoulder was abducted to 45° and the elbow flexed to 90°, with the dynamometer positioned just proximal to the wrist. In the assessments of shoulder flexion and extension using the MicroFET 2 Handheld Dynamometer, a slight flexion, approximately 5 to 10°, was intentionally allowed to accommodate natural movement tendencies and enhance participant comfort (*Biasini et al., 2023*). During strength testing, it is common for participants to instinctively bend the elbow slightly when exerting force against the dynamometer. This response helps stabilize the arm and prevent the elbow joint from hyperextending, which can occur under high force loads (*Biasini et al., 2023*). These testing positions are based on protocols validated by prior research to ensure accurate and reproducible measurements, aligning with biomechanical standards for isometric strength testing.

Each participant underwent a single data-collection session. Prior to commencing the testing, a comprehensive explanation of the procedure was provided to each participant, emphasizing the importance of exerting maximal effort. To reduce potential biases, the order of testing for each side (dominant and non-dominant) was randomized. Participants were allowed to complete 1 to 2 submaximal practice trials at each prescribed position to familiarize themselves with the task. A minimum of three trials was conducted for each assessed isometric function, with additional trials administered if the third trial exhibited greater strength than the first two. This approach served as a precautionary measure against potential motor learning effects. Trials were executed with approximately 5-second intervals between them, while 1-minute rests were observed between positions within the

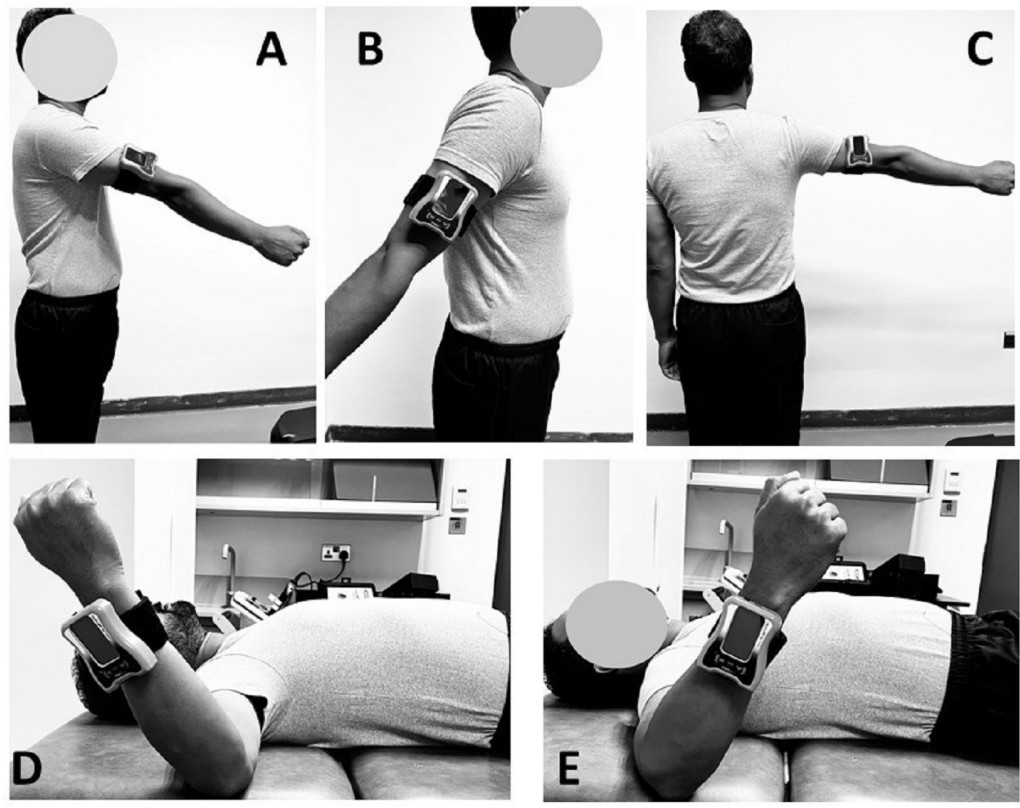

**Figure 2** Shoulder Joint reposition sense evaluation using digital inclinometer in (A) flexion; (B) extension; (C) abduction; (D) external rotation and (E) internal rotation.

same plane, and 2-minute intervals were enforced between functions in different planes. The entire testing session, inclusive of obtaining written informed consent, typically lasted between 30 and 45 min. The maximum strength achieved in Newton meters (Nm) from the three best trials was subsequently calculated and stored for later analysis on a personal computer.

## Shoulder Joint reposition sense evaluation

Shoulder joint reposition sense (JRS) evaluations were conducted with participants positioned either standing or in a supine lying position, following standardized protocols to ensure consistency and reliability across measurements (*Ager et al., 2017*; *Alfaya et al., 2023a*). For shoulder flexion, extension, and abduction, participants stood upright, and a digital inclinometer was securely attached to the lateral aspect of the arm to accurately assess these movements (Figs. 2A, 2B & 2C) (*Ager et al., 2017*; *Vafadar, Côté & Archambault, 2016*). For assessments involving external and internal rotation, participants were positioned supine on a couch with their arms abducted to 90 degrees and elbows bent to 90 degrees (Figs. 2D & 2E). This specific positioning aligns the forearm's movement

plane with gravity, facilitating accurate angle measurements by the digital inclinometer, a method supported by *Ager et al. (2017)* and *Vafadar, Côté & Archambault (2016)*. Participants were instructed to actively replicate designated shoulder angles—60 degrees for flexion, 30 degrees for extension, 60 degrees for abduction, and 45 degrees for both internal and external rotation, standardized across all participants to reflect positions commonly utilized in clinical assessments. These target angles were selected based on their relevance in clinical settings rather than the exact mid-range of each participant's total range of motion (*Alfaya et al., 2023a*; *Alfaya et al., 2023b*). The total range of motion (ROM) for each position was not measured, and the mid-range positions were not explicitly quantified for each participant. Instead, these target angles were chosen because they are commonly used in clinical practice to evaluate shoulder proprioception, even though literature suggests that proprioceptive accuracy may be highest near 90 degrees of flexion (*Hillier, Immink & Thewlis, 2015*). This selection method, while practical for clinical assessments, may not align with findings indicating highest proprioceptive acuity at different angles.

The examiner guided the participant's shoulder to the target position and maintained it for 3 s, allowing the participant to memorize the position accurately before returning the shoulder to the neutral position for replication. The accuracy of shoulder repositioning was measured in degrees, and three trials were conducted for each target in each of the movement directions to ensure the reliability of the data, with the average of the three trials used for subsequent analysis. Errors in joint repositioning were quantified as absolute errors, representing the absolute value of the deviation from the target position, which helps minimize the influence of potential outliers and provides a more reliable representation of shoulder JRS accuracy (*Alfaya et al., 2023b*). This meticulous approach to measuring joint position sense underscores our commitment to producing valid and actionable research findings.

During the evaluations, several measures were implemented to minimize examiner interference and ensure that feedback from the evaluator did not influence the participant's performance. The examiner's contact with the participant's limb was minimized to the essential touch required to position the digital inclinometer, reducing any proprioceptive feedback that might influence the participant's performance. Clear and standardized instructions were provided to the participants to ensure uniform testing conditions, and participants were blindfolded during the shoulder flexion, extension, and abduction tests to eliminate visual feedback. For internal and external rotation assessments, participants were placed in a standardized supine position to maintain consistent testing conditions. The examiner maintained a neutral stance and minimized any verbal or non-verbal cues that could provide feedback to the participants during the tests.

## Sample size estimation

In the design of our study, we utilized G*Power statistics to estimate the required sample size, basing our calculations on an effect size of 0.4, derived from previous studies on joint proprioception in diabetes (*Ettinger, Boucher & Simonovich, 2018*). This medium effect size, along with a standard power setting of 0.80 and an alpha level of 0.05, led us to a final sample size of 172 participants. This total includes 86 individuals diagnosed with type 2

diabetes mellitus (T2DM) and a comparably sized, age-matched asymptomatic group of 86 individuals.

## Data analysis

The statistical analysis of the collected data were conducted using the IBM SPSS Statistics software (version 26; SPSS Inc., Armonk, NY, USA). Prior to conducting the analysis, the data were assessed for normality of distribution using the Shapiro–Wilk test. The results indicated that the data for both shoulder strength and joint repositioning accuracy variables followed a normal distribution ($p > 0.05$), allowing for the application of parametric statistical tests. Descriptive statistics were used to summarize the demographic characteristics of the study participants. Mean and standard deviation (SD) were calculated for continuous variables. To assess and compare shoulder strength and joint repositioning accuracy between individuals with T2DM (DM) and asymptomatic individuals, independent t-tests were employed. To explore the correlation between shoulder strength and joint repositioning accuracy in individuals with type 2 DM, Pearson correlation coefficients were computed to determine the strength and direction of these relationships. The interpretation of the correlation coefficients followed the conventional guidelines: values close to +1 indicated a strong positive correlation, values close to -1 indicated a strong negative correlation and values close to 0 indicated no significant correlation. If $p$-values were less than 0.05, the results were considered statistically significant. Correlation coefficients (r) were interpreted as follows: $|r|<0.30$ indicated a very weak correlation, $0.31 \leq |r| <0.60$ indicated a moderate correlation, $|r|$ and $|r| \geq 0.61$ indicated a strong correlation. Significance levels were set at $p < 0.05$.

## RESULTS

The demographic analysis of the study cohort, comprising 86 individuals with T2DM and an equivalent number of asymptomatic subjects, revealed several significant findings (Table 2). Importantly, there were no statistically significant differences in mean age or gender distribution between the two groups ($p = 0.383$ and $p = 0.296$, respectively). However, notable disparities were evident in key metabolic and lifestyle factors. Specifically, individuals in the DM cohort exhibited a significantly higher mean body mass index (BMI) of $26.43 \pm 4.23$ and elevated HbA1c levels averaging $7.23 \pm 0.86$ compared to the asymptomatic group ($p < 0.001$ for both parameters). The duration of diabetes was relevant only to the DM cohort, with a mean duration of $6.5 \pm 3.2$ years. While the difference in smoking prevalence showed a slightly higher proportion of smokers (20.9%) in the DM cohort compared to the asymptomatic group (13.9%), it did not reach statistical significance ($p = 0.235$). Importantly, a substantial contrast was observed in physical activity levels, with the DM cohort reporting significantly fewer mean hours of physical activity per week ($2.15 \pm 1.56$) compared to the asymptomatic group ($3.92 \pm 2.28$, $p < 0.001$). Furthermore, a marked difference was noted in the prevalence of a family history of diabetes, with a significantly higher occurrence in the DM cohort (67.4%) relative to the asymptomatic group (37.2%), demonstrating statistical significance ($p < 0.001$).

**Table 2  Demographic characteristics of T2DM and asymptomatic subjects.**

| Characteristic | DM Group ($n = 86$) | Asymptomatic Group ($n = 86$) | $p$-value |
|---|---|---|---|
| Age (years) | 63.8 ± 8.32 | 62.99 ± 7.13 | 0.383 |
| -Mean ± SD | | | |
| Gender | | | 0.296 |
| - Male: n (%) | 46 (53.5%) | 40 (46.5%) | |
| - Female: n (%) | 40 (46.5%) | 46 (53.5%) | |
| Body Mass Index (BMI) | 26.43 ± 4.23 | 24.91 ± 3.64 | <0.001 |
| - Mean ± SD | | | |
| HbA1c Levels (%) | 7.23 ± 0.86 | 5.43 ± 0.46 | <0.001 |
| - Mean ± SD | | | |
| Duration of Diabetes (years) | 6.5 ± 3.2 | N/A | N/A |
| - Mean ± SD | | | |
| Smoking Status | | | 0.235 |
| - Smokers: n (%) | 18 (20.9%) | 12 (13.9%) | |
| - Non-Smokers: n (%) | 68 (79.1%) | 74 (86.1%) | |
| Physical Activity (hours/week) | 2.15 ± 1.56 | 3.92 ± 2.28 | <0.001 |
| - Mean ± SD | | | |
| Family History of Diabetes | | | <0.001 |
| - Yes: n (%) | 58 (67.4%) | 32 (37.2%) | |
| - No: n (%) | 28 (32.6%) | 54 (62.8%) | |

**Notes.**
Values are presented as mean ± standard deviation (SD) for continuous variables, and n (%) for categorical variables.

The results of the comparative analysis between individuals with T2DM and asymptomatic individuals, focusing on shoulder muscle strength and shoulder proprioception, are summarized in Table 3.

In terms of isometric muscle strength (measured in Newton-meters), significant differences were observed in all muscle groups and movements between the DM Group and the Asymptomatic Group. For flexion, both in the dominant and nondominant arms, the DM Group exhibited substantially higher mean values compared to the Asymptomatic Group, with mean differences of 10.44 and 8.67 Newton-meters, respectively ($p < 0.001$). Similarly, in extension, the DM Group displayed higher muscle strength in both arms, resulting in mean differences of 5.55 and 5.65 Newton-meters ($p < 0.001$). Abduction strength also demonstrated significant differences, with mean differences of 5.55 Newton-meters for the dominant arm and 1.33 Newton-meters for the non-dominant arm ($p < 0.001$). Additionally, external and internal rotation strength in both arms exhibited significant differences, with mean differences ranging from 3.36 to 6.12 Newton-meters ($p < 0.001$).

Shoulder JRS, measured in degrees (°), also revealed substantial differences between the DM and Asymptomatic Groups. In flexion and extension, both dominant and non-dominant arms of the DM Group exhibited significantly higher proprioception errors in comparison to the Asymptomatic Group, with mean differences ranging from 0.13 to 0.18 degrees ($p < 0.001$). Abduction, external rotation, and internal rotation also demonstrated

**Table 3 Comparisons of shoulder muscle strength and shoulder proprioception between DM and asymptomatic individuals.**

| | Variable | T2DM Group (n = 86) | Asymptomatic Group (n = 86) | Mean difference | Standard error | Cohens-d | p-value |
|---|---|---|---|---|---|---|---|
| Isomeric muscle strength (Newton-meters) | Flexion | | | | | | |
| | ●Dominant | 52.12 ± 8.63 | 41.68 ± 7.68 | 10.44 | 1.42 | 7.37 | <0.001 |
| | ●Nondominant | 50.23 ± 8.86 | 39.78 ± 8.67 | 8.67 | 1.32 | 6.54 | <0.001 |
| | Extension | | | | | | |
| | ●Dominant | 66.35 ± 10.34 | 59.73 ± 9.45 | 5.55 | 1.61 | 4.11 | <0.001 |
| | ●Nondominant | 64.36 ± 9.87 | 51.56 ± 8.98 | 5.65 | 1.71 | 4.34 | <0.001 |
| | Abduction | | | | | | |
| | ●Dominant | 41.23 ± 8.64 | 35.68 ± 7.45 | 5.55 | 1.33 | 4.17 | <0.001 |
| | ●Nondominant | 37.56 ± 9.87 | 32.78 ± 8.76 | | | | <0.001 |
| | External Rotation | | | | | | |
| | ●Dominant | 32.13 ± 6.78 | 28.74 ± 6.54 | 3.36 | 1.09 | 3.11 | <0.001 |
| | ●Nondominant | 29.56 ± 6,87 | 23.56 ± 7.86 | 3.56 | 1.45 | 3,87 | <0.001 |
| | Internal Rotation | | | | | | |
| | ●Dominant | 25.67 ± 5.67 | 20.23 ± 5.67 | 6.12 | 1.45 | 4.14 | <0.001 |
| | ●Nondominant | 21.65 ± 6.78 | 18.87 ± 6.21 | 5.34 | 1.34 | 3.67 | <0.001 |
| Shoulder Joint reposition sense (°) | Flexion | | | | | | |
| | ●Dominant | 4.35 ± 1.34 | 2.34 ± 0.78 | 1.01 | 0.13 | 3.67 | <0.001 |
| | ●Nondominant | 4.98 ± 1.34 | 2.56 ± 0.89 | 1.05 | 0.18 | 3.87 | <0.001 |
| | Extension | | | | | | |
| | ●Dominant | 5.34 ± 1.98 | 2.23 ± 0.45 | 2.07 | 0.18 | 4.56 | <0.001 |
| | ●Nondominant | 5.98 ± 1.23 | 2.56 ± 0.45 | 1.98 | 0.16 | 4.63 | <0.001 |
| | Abduction | | | | | | |
| | ●Dominant | 4.67 ± 1.10 | 1.36 ± 0.34 | 1.89 | 0.16 | 3.33 | <0.001 |
| | ●Nondominant | 4.98 ± 1.23 | 1.93 ± 0.55 | 1.78 | 0.15 | 3.87 | <0.001 |
| | External rotation | | | | | | |
| | ●Dominant | 4.76 ± 0.98 | 1.10 ± 0.25 | 1.89 | 0.13 | 3.66 | <0.001 |
| | ●Nondominant | 5.36 ± 1.45 | 1.45 ± 0.34 | 2.01 | 0.14 | 3.26 | <0.001 |
| | Internal rotation | | | | | | |
| | ●Dominant | 4.56 ± 1.23 | 2.01 ± 0.33 | 1.89 | 0.15 | 2.89 | <0.001 |
| | ●Nondominant | 4.87 ± 1.34 | 2.56 ± 0.59 | 1.71 | 0.16 | 3.12 | <0.001 |

**Notes.**

T2DM, Type 2 diabetes mellitus; D, dominant; ND, non-dominant.

significant proprioception differences, with mean differences ranging from 0.13 to 0.18 degrees ($p < 0.001$).

The effect sizes (Cohen's d) for all comparisons were notably large, indicating substantial differences between the DM Group and the asymptomatic Group in both muscle strength and proprioception. These findings suggest that individuals with T2DM exhibit impairments in shoulder muscle strength and proprioception compared to their
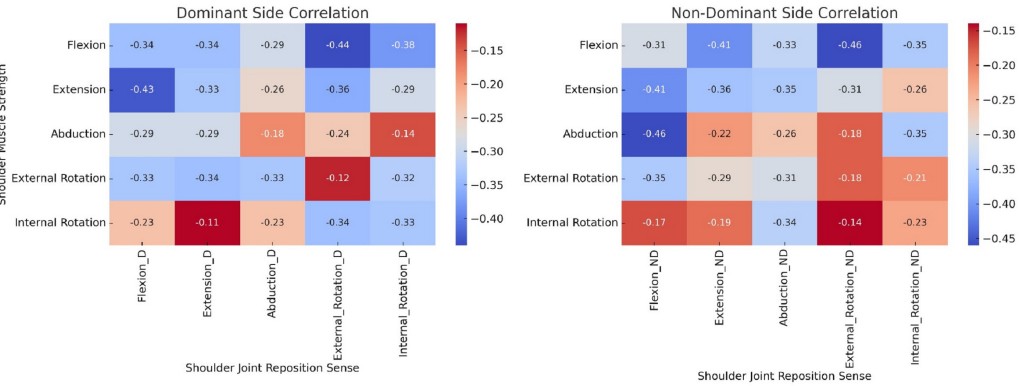

**Figure 3 Heat map of correlation between isometric muscle strength and joint reposition sense in individuals with diabetes mellitus.** Heat map visualizes the correlation coefficients between shoulder muscle strength (measured in Newton-meters) and shoulder joint reposition sense (measured in degrees) in a cohort of individuals with diabetes mellitus. The muscle strength parameters include flexion, extension, abduction, external rotation, and internal rotation, assessed for both the dominant and non-dominant sides. The color gradient represents the strength and direction of the correlation with warmer colors (red) indicating negative correlations.

asymptomatic counterparts, which may have implications for their musculoskeletal function and overall health.

The results of the correlation analysis between isometric shoulder muscle strength (Newton-meters), and shoulder JRS (°), in individuals with diabetes mellitus are summarized in Fig. 3. In the context of shoulder flexion, whether in the dominant or non-dominant arms, statistically significant negative correlations were observed. For the dominant arm, the correlation coefficients ranged from −0.29 to −0.46, while for the non-dominant arm, the range was from −0.26 to −0.41, with all correlations showing a $p$-value of less than 0.001. These findings imply that an increase in shoulder muscle strength is associated with a concomitant decrease in joint reposition errors during flexion. Akin to the findings in flexion, negative correlations were also evident in shoulder extension for both the dominant and non-dominant arms. In the dominant arm, correlation coefficients ranged from −0.18 to −0.29, while in the non-dominant arm, they ranged from −0.13 to −0.46, all with $p$-values less than 0.001. This implies that higher levels of muscle strength correspond to reduced errors in joint repositioning during extension. Consistently, in the case of abduction, external rotation, and internal rotation, negative correlations were established for both the dominant and non-dominant arms. The correlation coefficients ranged from −0.12 to −0.48 for the dominant arm and from −0.15 to −0.49 for the non-dominant arm, all with $p$-values below 0.001.

## DISCUSSION

The study had two main objectives: first, to evaluate and compare shoulder muscle strength and joint repositioning accuracy between individuals with T2DM and asymptomatic individuals; second, to investigate the correlation between shoulder strength and joint

repositioning accuracy in those with T2DM. The results demonstrated noteworthy reductions in both shoulder muscle strength and joint repositioning accuracy in individuals with T2DM compared to their asymptomatic counterparts. Additionally, a significant negative correlation was observed between shoulder muscle strength and shoulder joint position sense errors in individuals with T2DM.

Individuals diagnosed with T2DM demonstrated significant reductions in shoulder muscle strength across various muscle groups and movements when compared to their asymptomatic counterparts. This decrease in muscle strength can be attributed to several factors commonly associated with diabetes, including microvascular complications, neuropathy, and muscular atrophy (*Bianchi & Volpato, 2016*; *Parasoglou, Rao & Slade, 2017*). Additionally, individuals with diabetes often exhibit lower levels of physical activity and are more likely to engage in lifestyle practices that may contribute to this observed muscular weakness (*Bianchi & Volpato, 2016*; *Palermi et al., 2021*). In addition to diminished muscle strength, our study revealed that diabetic individuals also exhibited heightened errors in shoulder joint position sense. This decline in proprioceptive abilities is frequently linked to diabetic neuropathy, which negatively affects the sensory nerves responsible for accurately perceiving joint positions (*Ahmad et al., 2019*; *Ettinger, Boucher & Simonovich, 2018*; *Gulley Cox, 2020*). Impaired proprioception can have significant consequences, including disruptions in balance, coordination, and overall functional abilities (*Goble et al., 2009*; *Sturnieks, George & Lord, 2008*). The substantial discrepancies in both muscle strength and proprioception, underscored by notable effect sizes (Cohen's d), underscore the critical need for a comprehensive approach to musculoskeletal assessment and management within the diabetic population (*Ahmad et al., 2023*).

Our study's findings align with those of previous researchers in the field. *Savaş et al. (2007)* conducted a study investigating individuals with diabetes, reporting significantly lower muscle strength when compared to non-diabetic individuals (*Savaş et al., 2007*). They attributed this reduction in strength to factors such as diabetic neuropathy and muscle atrophy, findings that mirror our observations (*Savaş et al., 2007*). Similarly, *Gulley Cox (2020)* explored joint position sense in patients with T2DM, uncovering substantial deficits, particularly within the upper extremity (*Gulley Cox, 2020*). These outcomes corroborate our study's findings, which also highlighted proprioceptive impairments among individuals with diabetes (*Gulley Cox, 2020*). *Sözen et al. (2018)* conducted a comprehensive analysis of musculoskeletal function in diabetic patients, identifying a significant decrease in both muscle strength and proprioception (*Sözen et al., 2018*). Their research linked these impairments with elevated HbA1c levels and longer durations of diabetes, further reinforcing the results of our study (*Luo et al., 2018*). Moreover, a systematic review conducted by *Hillier, Immink & Thewlis (2015)* and *Mohamed & Jan (2020)* synthesized evidence from various studies and concluded that diabetes is consistently associated with reduced muscle strength and proprioceptive deficits across different joints, including the shoulder (*Hillier, Immink & Thewlis, 2015*; *Mohamed & Jan, 2020*). These collective findings provide robust support for our own, emphasizing the persistent presence of reduced shoulder muscle strength and impaired proprioception in individuals with T2DM compared to asymptomatic individuals (*Mohamed & Jan, 2020*). Taken together, these

results underscore the importance of implementing targeted interventions to address these musculoskeletal issues within the diabetic population, ultimately leading to improved overall health and quality of life (*Izquierdo et al., 2021*; *Onder et al., 2015*).

The observed negative correlations between isometric shoulder muscle strength and JRS in individuals with T2DM provide valuable insights into the complex relationship between musculoskeletal function and metabolic health. These correlations can be attributed to several plausible mechanisms. Firstly, diabetic neuropathy, a well-documented complication of T2DM, plays a significant role (*Leung & Lam, 2000*). Prolonged exposure to high blood sugar levels can damage sensory and motor nerves responsible for muscle control and proprioception, leading to muscle weakness and impaired joint position sense (*Proske & Gandevia 2012*; *Reeves, Orlando & Brown, 2021*; *Zochodne & Toth, 2014*). This aligns with studies conducted by *Andersen et al. (1998)* and *Oh et al. (2019)*, which has highlighted the association between diabetic neuropathy and muscle weakness. Secondly, prolonged hyperglycemia in T2DM can negatively affect muscle tissue itself (*Lewis et al., 2002*). Chronic high blood sugar levels may lead to muscle atrophy and reduced muscle quality, resulting in decreased muscle strength (*Johansen et al., 2003*). Additionally, the accumulation of advanced glycation end products (AGEs) in muscle tissues due to hyperglycemia can contribute to muscle stiffness and reduced joint flexibility (*Haus et al., 2007*). Studies by *Mapanga & Essop (2016)* and *Dial et al. (2021)* has demonstrated these adverse effects of hyperglycemia on muscle structure and function. Furthermore, physical inactivity, often prevalent in individuals with T2DM, can exacerbate these issues (*Kanaley et al., 2022*). Reduced physical activity levels can lead to muscle deconditioning and loss of muscle mass, further impairing muscle strength (*Kanaley et al., 2022*). The studies by *Kanaley et al. (2022)* (2015) and *Amanat et al. (2020)* have highlighted the detrimental impact of physical inactivity on muscle function and strength in individuals with diabetes, which supports our findings of a negative correlation between muscle strength and JRS (*Amanat et al., 2020*).

The clinical significance of this study becomes evident when considering the multifaceted impact of T2DM on musculoskeletal function and proprioception (*Tan, 2020*). Reduced shoulder muscle strength, attributed to factors like diabetic neuropathy, muscular atrophy, and lifestyle-related factors, has implications for functional limitations, quality of life, and susceptibility to musculoskeletal injuries (*D'Onofrio et al., 2023*). Moreover, deficits in shoulder joint position sense (proprioception), often linked to diabetic neuropathy, can compromise balance, coordination, and overall physical functioning (*D'Onofrio et al., 2023*). The negative correlations between muscle strength and joint proprioception underscore the intricate interplay between metabolic health and musculoskeletal function (*Izquierdo et al., 2021*). This highlights the importance of early detection of diabetic neuropathy, glycemic control, and tailored exercise interventions in diabetes management (*Izquierdo et al., 2021*).

While providing valuable insights, this study has several limitations. Its cross-sectional design hampers establishing causation, emphasizing the need for longitudinal research. The participant pool primarily comprised asymptomatic individuals and those with T2DM, potentially excluding varying degrees of diabetes-related complications and

comorbidities. Diversifying the sample would enhance findings' comprehensiveness. It's important to consider T2DM's broader impact on musculoskeletal parameters beyond this study's scope. Another limitation is the absence of investigating interventions to mitigate musculoskeletal impairments in T2DM, a topic for future research. Relying solely on self-reported physical activity levels introduces limitations, emphasizing the need for objective measures like accelerometry. While absolute error measures overall error magnitude, it doesn't differentiate between overshooting and undershooting targets, crucial for proprioceptive acuity assessment. Future studies may incorporate constant error and variable error metrics for a detailed understanding. Additionally, future research should explore potential mediators or moderators of observed relationships, including glycemic control, neuropathy severity, and lifestyle factors, to understand the musculoskeletal implications of T2DM better.

## CONCLUSIONS

This study conclusively demonstrates the pronounced impact of Type 2 diabetes mellitus (T2DM) on shoulder muscle strength and proprioception. Our results indicate significant reductions in shoulder muscle strength and joint repositioning sense (JRS) in individuals with T2DM compared to asymptomatic controls. These deficits significantly affect physical function, increasing the risk of falls and musculoskeletal injuries, thereby highlighting the importance of including musculoskeletal evaluations in the clinical management of T2DM. Notably, our findings of negative correlations between muscle strength and JRS illustrate the complex interactions between muscular and proprioceptive functions in diabetic individuals. The study underscores the need for early detection of diabetic neuropathy and tailored interventions aimed at mitigating its adverse effects on muscle strength and proprioception.

## ACKNOWLEDGEMENTS

The authors express their gratitude to all the participants who participated in this study.

### Funding

The Deanship of Research and Graduate Studies at King Khalid University funded this work through the Large Research Project under grant number RGP2/23/45. The funders had no role in study design, data collection and analysis, decision to publish, or preparation of the manuscript.

### Grant Disclosures

The following grant information was disclosed by the authors:
The Deanship of Research and Graduate Studies at King Khalid University: RGP2/23/45.

### Competing Interests

The authors declare there are no competing interests.

## Author Contributions

- Ravi Shankar Reddy conceived and designed the experiments, performed the experiments, analyzed the data, prepared figures and/or tables, authored or reviewed drafts of the article, and approved the final draft.
- Mastour Saeed Alshahrani conceived and designed the experiments, performed the experiments, analyzed the data, prepared figures and/or tables, authored or reviewed drafts of the article, and approved the final draft.
- Mohammad A. ALMohiza conceived and designed the experiments, performed the experiments, prepared figures and/or tables, authored or reviewed drafts of the article, and approved the final draft.
- Batool Abdulelah Alkhamis conceived and designed the experiments, performed the experiments, analyzed the data, authored or reviewed drafts of the article, and approved the final draft.
- Jaya Shanker Tedla conceived and designed the experiments, performed the experiments, analyzed the data, authored or reviewed drafts of the article, and approved the final draft.
- Venkata Nagaraj Kakaraparthi conceived and designed the experiments, performed the experiments, authored or reviewed drafts of the article, and approved the final draft.
- Ghada Mohamed Koura performed the experiments, authored or reviewed drafts of the article, and approved the final draft.
- Debjani Mukherjee performed the experiments, authored or reviewed drafts of the article, and approved the final draft.
- Hani Hassan Alnakhli performed the experiments, authored or reviewed drafts of the article, and approved the final draft.
- Hussain Saleh H. Ghulam performed the experiments, authored or reviewed drafts of the article, and approved the final draft.
- Raee S. Alqhtani performed the experiments, authored or reviewed drafts of the article, and approved the final draft.

## Human Ethics

The following information was supplied relating to ethical approvals (*i.e.*, approving body and any reference numbers):

King Khalid University Ethics Committee

## Data Availability

The raw measurements are available in Supplementary Files 1

## Supplemental Information

Supplemental information for this article can be found online at http://dx.doi.org/10.7717/peerj.17630#supplemental-information.

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
