# Peer review of "Shoulder muscle weakness and proprioceptive impairments in type 2 diabetes mellitus: exploring correlations for improved clinical management"

_PeerJ, doi:10.7717/peerj.17630_

## Round 0.1 · original submission · Major Revisions

The manuscript is interesting, but there are some methodological aspects that should be addressed:

- please provide references that sustain the reliability of isometric muscle strength assessment. The reference the authors mentioned in the text (Saccol et al 2017) refers only to the reliability of shoulder rotators assessment. Moreover, the position used in the present study to assess the shoulder rotators are not those described in the reference.

- please specify the muscles that were tested for each movement and also described the testing positions (with references for the testing positions). Also explain the position of the dynamometer.

- please explain why have the authors chose to test the internal and external rotators from 45 degrees of abduction?

- for the flexion and extensors muscles strength assessment why was the elbow flexed (as it can be seen in Figure 1)?

- please provide references for the validity and reliability of the joint position sense evaluation. How and why those specific angles were chosen?

- the conclusion section should be more concise.

Reviewer 1 ·

Basic reporting

Multiple instances of inappropriate language with respect to data. Data is a plural noun, where the correct usage with respect to data should be data were, or data are, not data was.

Experimental design

Under the shoulder joint reposition sense (JRS) evaluation, it is unclear from the writing how many trials were performed for each target for each position (flexion, extension, abduction and rotation). The calculation of errors in degrees (dependent variable) associated with these evaluations is unclear. If the average error were quantified, it should be reported as the constant error; however, many studies have indicated that due to overshooting and undershooting targets, the constant error can be a poor representation of the actual proprioceptive acuity. Suprak et al., 2006 report Absolute error (using the absolute value of error in quantification of the average) for example; however, this metric loses the fidelity of the direction of the error (overshoot versus undershoot). Alternatively, studies by Ettinger et al., 2018 have indicated that a combination of constant error and variable error are needed because they capture both the accuracy and precision associated with joint angle replication while maintaining the fidelity of overshoot and undershooting the target. A better description of these errors and how they were evaluated is needed in order to make inferences on reported data.

The measurement of joint angles using the digital inclinometer poses several concerns that could be addressed in the body of the manuscript. For one, it is unclear how each unique shoulder movement could be quantified with a digital inclinometer since these devices rely on the vector of gravity to establish joint angle. Both flexion and extension, abduction and adduction are fairly easy to conceptualize with respect to gravity but should still be described in text. Both internal and external rotation of the shoulder would be difficult to measure with respect. to gravity unless the arm were abducted to 90 degrees and the elbow bent to 90 degrees; which appears consistent with Figure 2 pictures D and E; however there is no description of these positions in the text to confirm.

The target angles chosen to quantify JRS is a bit confusing in how they were selected. It is not apparent if the JRS targets for flexion, extension, abduction and rotation were selected based on a percentage of range of motion (and what percent that was) or if they were standardized target angles that all participants must replicate independently from what their total range of motion was. We know that there is a strong influence of joint angle on repositioning error; where most joints appear to have the highest acuity nearest 90 degrees (Suprak 2006). Due to this influence, many authors have chosen the standardized target angle approach to "wash-out" error discrepancies that could occur between-subject due to being closer to the 90 degree mark of joints (which is more likely to happen when target angles are selected based on the total range of motion of the participant).

Validity of the findings

The impact of this work is important in our understanding of T2DM and shoulder health. More attention to addressing the methodological concerns highlighted above will only serve to increase our confidence in these findings.

Additional comments

I believe the authors should additionally cite the following articles due to their novelty in the areas of both proprioception and measuring proprioceptive errors in the T2DM populations:

Ettinger et al., (2018) Patients with type 2 diabetes demonstrate proprioceptive deficit in the knee. World Journal of Diabetes. 9(3):59-65.

Suprak et al., (2006) Shoulder Joint Position Sense Improves with Elevation Angle in a Novel, Unconstrained Task. Journal of Orthopedic Research.

Reviewer 2 ·

Basic reporting

No comment

Experimental design

The explanation about methodology needs more detail. I suggest that the authors improve the description to provide more explanation on sampling method, how did recruit the participants, and data collection process.

I suggest that the authors include reference(s) for the measurement tool used for the outcome on Joint Reposition Sense.

Validity of the findings

The conclusion (lines 381-390) needs to be more concise and direct in addressing the study objectives. I suggest that the authors improve the conclusion by focusing on what they have found in the study, aligning it with the study objectives.

Additional comments

Please check the spacing of the paragraph in sample size estimation section.
Please check the format of the citations in paragraph 3 and 4 of the discussion section.

Annotated reviews are not available for download in order to protect the identity of reviewers who chose to remain anonymous.

---

## Round 0.2 · Minor Revisions

Thank you for your revisions however Reviewer 1 has minor revisions required to your manuscript. Please address the 3 points. Thanks, A/Prof M Climstein

Reviewer 1 ·

Basic reporting

This is much improved. The writing is more clear and concise.

Experimental design

I still have three methodological concerns with the study in its present form.
1. Previously I had commented that The measurement of joint angles using the digital inclinometer poses several concerns that could be addressed in the body of the manuscript. For one, it is unclear how each unique shoulder movement could be quantified with a digital inclinometer since these devices rely on the vector of gravity to establish joint angle. Both flexion and extension, abduction and adduction are fairly easy to conceptualize with respect to gravity but should still be described in text. Both internal and external rotation of the shoulder would be difficult to measure with respect. to gravity unless the arm were abducted to 90 degrees and the elbow bent to 90 degrees; which appears consistent with Figure 2 pictures D and E; however there is no description of these positions in the text to confirm. In the revised manuscript (Line 204) the authors cite that the methods used for arm angle measurement was methodologically supported by the 2006 Suprak study. However, this is not true, as the 2006 Suprak study did not use a digital inclinometer, instead they used three dimensional electromagnetic tracking system to quantify scapular and humeral kinematics at 40 Hz.
2. Line 213: The authors indicate that the target angles represents the “Mid-range position” of range of motion, however; there is no mention of what the total range of motion was for a particular position. Were any measurements of the participant total ROM made in each of these positions? Were mid-range position quantified? Ultimately, it is unclear how these positions were established. The authors cite the 2023 study by Alfaya et al., however, this is the same research group conducting the current study so there is no external verification that this method works. Furthermore, stating that proprioceptive accuracy is greater in the midpoint of range of motion (Line 216) is at odds with the majority of proprioceptive literature, which suggests that the highest acuity is actually near 90 degrees of flexion across multiple joints including the shoulder joint (same Suprak 2006 article from earlier). I think it would be helpful to indicate that the target angles were selected near the mid-range position which represents the common clinical evaluative position.
3. There should be mention of what steps were taken to avoid examiner interference on the participant during proprioceptive testing, since the evaluator is making contact with the proprioceptive limb with the digital inclinometer. Is it possible that feedback from the evaluator is passed to the subject through this contact?

Validity of the findings

See previous comment.

Reviewer 2 ·

Basic reporting

no comment

Experimental design

I agree with all the amendments have been made.

Validity of the findings

I agree with all the amendments have been made.

Additional comments

None

---

## Round 0.3 · accepted · Accept

Thank you for addressing Reviewer 2 comments with regard to the manuscript. I am pleased to recommend your amended manuscript for publication. We look forward to receiving future manuscripts from you and your colleagues.